# The CD2 isoform of protocadherin-15 is an essential component of the tip-link complex in mature auditory hair cells

Elise Pepermans[1,2,3], Vincent Michel[1,2,3], Richard Goodyear[4], Crystel Bonnet[2,3,5], Samia Abdi[6], Typhaine Dupont[1,2,3], Souad Gherbi[7], Muriel Holder[8], Mohamed Makrelouf[9], Jean-Pierre Hardelin[1,2,3], Sandrine Marlin[7], Akila Zenati[9], Guy Richardson[4], Paul Avan[10,11,12], Amel Bahloul[1,2,3] & Christine Petit[1,2,3,5,13,*]

## Abstract

**Protocadherin-15 (Pcdh15) is a component of the tip-links, the extracellular filaments that gate hair cell mechano-electrical transduction channels in the inner ear. There are three Pcdh15 splice isoforms (CD1, CD2 and CD3), which only differ by their cytoplasmic domains; they are thought to function redundantly in mechano-electrical transduction during hair-bundle development, but whether any of these isoforms composes the tip-link in mature hair cells remains unknown. By immunolabelling and both morphological and electrophysiological analyses of post-natal hair cell-specific conditional knockout mice (*Pcdh15*[ex38-fl/ex38-fl] *Myo15-cre*[+/−]) that lose only this isoform after normal hair-bundle development, we show that Pcdh15-CD2 is an essential component of tip-links in mature auditory hair cells. The finding, in the homozygous or compound heterozygous state, of a *PCDH15* frameshift mutation (p.P1515Tfs*4) that affects only Pcdh15-CD2, in profoundly deaf children from two unrelated families, extends this conclusion to humans. These results provide key information for identification of new components of the mature auditory mechano-electrical transduction machinery. This will also serve as a basis for the development of gene therapy for deafness caused by *PCDH15* defects.**

**Keywords** auditory mechano-electrical transduction; deafness; protocadherin-15; stereocilia; tip-link

**Subject Categories** Genetics, Gene Therapy & Genetic Disease; Neuroscience

## Introduction

Three transmembrane protocadherin-15 (Pcdh15) splice isoforms (Pcdh15-CD1, Pcdh15-CD2 and Pcdh15-CD3) differing only in the C-terminal part of their cytoplasmic domains (Fig 1A) are present in the hair bundles of developing cochlear hair cells (Ahmed *et al*, 2006). The inner and outer hair cells (IHCs and OHCs) of the cochlea have distinct roles (signal transmission for IHCs; frequency dependent mechanical amplification for OHCs), but both perform mechano-electrical transduction (MET). MET takes place in the hair bundle, an apical ensemble of stiff microvilli (stereocilia) organized in three rows of increasing height, the short, middle and tall rows. The oblique tip-link connects the tip of a stereocilium in one row to the side of an adjacent taller stereocilium. This link controls the open probability of the MET channels located at its lower insertion point, namely at the tips of short- and middle-row stereocilia (Howard & Hudspeth, 1988; Beurg *et al*, 2009). Pcdh15 and cadherin-23 form the lower and upper parts of this link, respectively (Kazmierczak *et al*, 2007). Kinociliary links, also composed of these cadherins (Goodyear *et al*, 2010), connect the stereocilia to the kinocilium, a structure that regresses before the onset of hearing but is necessary for the correct planar polarization of the hair bundle during the early stages of development.

Auditory defects are not detected in mice lacking Pcdh15-CD1 or Pcdh15-CD3, whereas mice lacking Pcdh15-CD2 (*PCDH15-ΔCD2*) are profoundly deaf. However, tip-links are observed and MET currents can be recorded in immature hair cells of *PCDH15-ΔCD2* mice, and consequently, it has been suggested that the three Pcdh15 isoforms function redundantly in the tip-link (Webb *et al*, 2011).

1. Unité de Génétique et Physiologie de l'Audition, Institut Pasteur, Paris, France
2. UMRS 1120, Institut National de la Santé et de la Recherche Médicale (INSERM), Paris, France
3. Université Pierre et Marie Curie (Paris VI), Paris, France
4. School of Life Sciences, University of Sussex, Brighton, UK
5. Syndrome de Usher et autres Atteintes Rétino-Cochléaires, Institut de la vision, Paris, France
6. Centre Hospitalier universitaire de Blida, Université Saad Dahleb, Blida, Algérie
7. Centre de référence des Surdités Génétiques, Hôpital Necker, Paris, France
8. Service de Génétique Clinique, Hôpital Jeanne-de-Flandre, Lille, France
9. Laboratoire de Biochimie Génétique, Université d'Alger 1, Alger, Algérie
10. Laboratoire de Biophysique Sensorielle, Université d'Auvergne, Clermont-Ferrand, France
11. UMR 1107, Institut National de la Santé et de la Recherche Médicale (INSERM), Clermont-Ferrand, France
12. Centre Jean Perrin, Clermont-Ferrand Cedex 01, France
13. Collège de France, Paris, France
*Corresponding author. Tel: +33 145688890; E-mail: cpetit@pasteur.fr

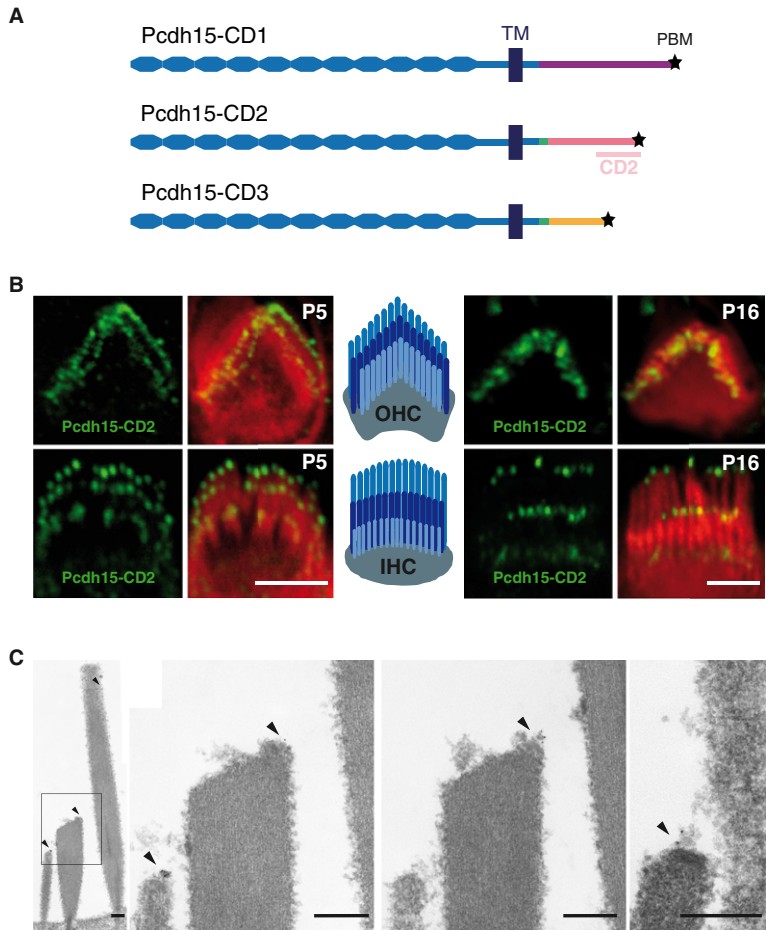

**Figure 1. Immunolabelling of Pcdh15-CD2.**

A   Schematic of Pcdh15 isoforms. The position of the CD2 fragment used to produce the anti-Pcdh15-CD2 antibody is indicated (138 C-terminal amino acids). (TM: transmembrane domain, * PBM: PDZ-binding motif).

B   Confocal images of hair cells (OHC above, IHC below) stained for Pcdh15-CD2 (green) and actin (red) at P5 (immature hair cells) and P16 (mature hair cells). Scale bars: 2 μm.

C   Transmission electron micrographs of a Pcdh15-CD2 immunoreactive IHC hair bundle at P15 (left panel), enlargements of boxed region: two adjacent sections of the same hair bundle (central panels) and a tip-link profile from an OHC (right panel). Arrowheads indicate gold particles. Note that the presence of the gold particles is restricted to the tips of the short- and middle-row stereocilia, consistent with Pcdh15-CD2 being a component of the lower part of the tip-link, and to the apico-lateral region of the tallest stereocilia, suggesting that Pcdh15-CD2 could also be a component of the lateral links between stereocilia of the tallest row. Scale bars: 200 nm.

The deafness of *PCDH15-ΔCD2* mice has been attributed to defects in hair-bundle polarity, a phenotype consistent with Pcdh15-CD2 being an essential component of kinociliary links (Ahmed *et al*, 2006; Webb *et al*, 2011). Considering that reported mouse mutants with misoriented hair bundles display only moderate hearing impairments (Curtin *et al*, 2003; Jagger *et al*, 2011; Copley *et al*, 2013), we investigated whether Pcdh15-CD2 could play a previously unrecognized but critical role in mature auditory hair cells.

# Results and Discussion

## Pcdh15-CD2 is located at the stereociliary tips in auditory hair bundles

Previous immunofluorescence studies have not demonstrated that Pcdh15-CD2 is present in mature hair bundles (Ahmed *et al*, 2006).

We therefore generated a new polyclonal antibody specific to this isoform (Supplementary Fig S1): from post-natal day 5 (P5) onwards, Pcdh15-CD2 was detected at the tip of every stereocilium, in both IHCs and OHCs (Fig 1B). In mature IHCs, Pcdh15-CD2 antibody labelling was conspicuous at the tips of the three rows of stereocilia, in particular at the lower tip-link insertion points. Transmission electron microscopy of immunogold-labelled longitudinal sections of hair bundles showed that almost all gold particles were at the apices of the three stereociliary rows in mature OHCs and IHCs (Fig 1C). The high concentration of gold particles at the extreme apex of a small row IHC stereocilium in one section (Fig 1C, second panel) and at the extreme apex of a neighbouring, middle-row stereocilium in an adjacent section (Fig 1C, third panel) exemplifies the restricted distribution of Pcdh15-CD2. A typical tip-link profile from an OHC hair bundle showing immunogold labelling at the tip-link lower insertion point is also shown in the last panel of Fig 1C. These observations suggest that Pcdh15-CD2 is a lower

tip-link component in mature auditory hair cells (see also Supplementary Figs S2 and S3).

## Absence of Pcdh15-CD2 results in the loss of tip-links in mature auditory hair cells

To probe the role of Pcdh15-CD2 in mature hair bundles, a post-natal hair cell-specific conditional knockout mouse model, $Pcdh15^{ex38-fl/ex38-fl}Myo15-cre^{+/-}$ mice, was generated. Conditional post-natal deletion of exon 38, specific to the Pcdh15-CD2 isoform, circumvented the early morphogenetic defects caused by the absence of this isoform during hair-bundle development (Webb et al, 2011; see Methods and Supplementary Fig S4). The auditory function of these mutant mice was probed by in vivo audiometric tests, which explore the activities of IHCs and OHCs. At the onset of hearing, on P15, auditory function, measured as auditory brainstem responses (ABRs), was identical in $Pcdh15^{ex38-fl/ex38-fl}Myo15-cre^{+/-}$ mice (referred to as conditional Pcdh15Δ'CD2 mice) and their $Pcdh15^{ex38-fl/ex38-fl}$ littermate controls. By P17, ABR thresholds in the mutants started to increase, and by P30, they were above 90 dB SPL across the frequency spectrum tested (5–40 kHz). By P45, the conditional Pcdh15Δ'CD2 mice lacked any identifiable ABR response to loud sound stimulation (115 dB SPL), indicating complete hearing loss and fully defective IHCs. Distortion-product otoacoustic emissions (DPOAEs), which involve OHC MET channel function (Avan et al, 2013), increased in threshold and decreased in amplitude from P24 onwards, had almost disappeared by P30 and were completely absent on P45. Cochlear microphonic (CM) potentials, phasic extracellular potentials reflecting MET currents in the OHCs of the basal region of the cochlea (Patuzzi et al, 1989), had an amplitude reduced to 4% of that in controls by P30, indicating a loss of MET in OHCs. As ABR thresholds were 40 dB higher on P21 despite normal DPOAEs, this indicates that IHC function was already impaired at this age. Consistent with this, the amplitude of compound action potentials [representing synchronous firing of afferent neurons innervating the IHCs (Spoendlin & Baumgartner, 1977)] in response to loud sound stimuli (105 dB SPL, the processing of which relies only on IHC function) on P30 was only 3% of that in controls (Fig 2A–C and Supplementary Fig S5). Thus, MET, whilst initially normal in both IHCs and OHCs in conditional Pcdh15Δ'CD2 mice, is totally abolished by P45. In contrast, conditional Pcdh15Δ'CD2 mice explored by behavioural tests (see Methods) did not show vestibular dysfunction, as is the case for PCDH15-ΔCD2 mice (Webb et al, 2011).

Auditory hair bundles were analysed morphologically by scanning electron microscopy (Fig 2D and E). In conditional Pcdh15Δ'CD2 mice, hair bundles were correctly oriented, unlike those of $Pcdh15^{ex38-fl/ex38-fl}$ $PGK-cre^{+/-}$ mice (referred to as KO Pcdh15Δ'CD2 mice) that lack Pcdh15-CD2 throughout development (Fig 2D; see Materials and Methods and Supplementary Figs S4 and S6). In conditional Pcdh15Δ'CD2 mice, very few tip-links were observed on OHCs on P30, and none were detected by P45; these links were consistently observed in littermate controls at the same ages. In P30 IHCs of conditional Pcdh15Δ'CD2 mice, most middle-row stereocilia had lost their distal prolate shape, indicating the loss of tip-link tension (Tilney et al, 1988). By P45, many middle-row stereocilia of IHCs had reduced lengths and most, if not all, short-row stereocilia had regressed entirely (Fig 2E). Some OHC short-row stereocilia were also missing. These anomalies are reminiscent of those reported in post-natal conditional knockout mice lacking other proteins of the tip-link complex (Caberlotto et al, 2011) and consequently are consistent with the existence of a functional connection between the tip-link and F-actin polymerization in the stereocilia.

Thus, in conditional Pcdh15Δ'CD2 mice, when Pcdh15-CD2 is no longer expressed in auditory hair bundles, tip-links are lost, both from IHCs and OHCs: this explains the loss of MET. These results demonstrate that Pcdh15-CD2 is an essential component of tip-links in mature auditory hair cells, which accounts for the profound deafness of mutant mice lacking this isoform.

## Patients lacking PCDH15-CD2 are profoundly deaf

In humans, biallelic loss-of-function mutations in PCDH15 result in Usher syndrome of type 1 (Usher 1), a dual sensory disorder combining severe to profound congenital deafness, vestibular disorders and prepubertal onset retinitis pigmentosa eventually leading to blindness. To date, no Usher 1 patient carrying a mutation specifically affecting only one of the three Pcdh15 isoforms has been reported, although more than 45 different PCDH15 mutations have been detected in the about 400 patients analysed (https://grenada.lumc.nl/LOVD2/Usher_montpellier/USHbases.html; Bonnet et al, 2011; and Crystel Bonnet, unpublished results). We therefore extended our search for specific Pcdh15 isoform defects to patients affected by isolated (nonsyndromic) deafness. Sanger sequencing was used to analyse PCDH15 in 60 unrelated individuals with congenital profound sensorineural deafness, for whom common pathogenic mutations in the most prevalent deafness genes (GJB2, MYO15A and OTOF) had been excluded. In three patients from two independent families (patient IV.2 from family CPID4744 and patients IV.4 and IV.5 from family CPIDS6-10), we found a frameshift mutation specific to the CD2 isoform: c.4542dup (p.P1515Tfs*4), in exon 38 of PCDH15. This mutation was not found in the Exome Variant Server database (http://evs.gs.washington.edu/EVS/). It is predicted to lead either to a

**Figure 2.  Auditory testing and morphological analysis of hair bundles in conditional Pcdh15Δ'CD2 mice.**

A    ABR thresholds across the 5-40 kHz frequency spectrum on P30 and P45. Blue and red curves (mean ± SEM) correspond to $Pcdh15^{ex38-fl/ex38-fl}$ (control) and conditional Pcdh15-Δ'CD2 mice, respectively. In P45 conditional Pcdh15Δ'CD2 mice, ABR waves could not be detected even at 115 dB SPL (P30: conditional Pcdh15Δ'CD2 mice n = 7, control n = 15, P45: conditional Pcdh15Δ'CD2 mice n = 6, control n = 15).

B, C   CM and CAP responses to a 10 kHz, 105 dB SPL tone burst in a $Pcdh15^{ex38-fl/ex38-fl}$ P30 control mouse (blue) and a P30 conditional Pcdh15Δ'CD2 mouse (red).

D    Scanning electron micrographs showing that the orientation of OHC hair bundles is normal in P30 conditional Pcdh15Δ'CD2 mice in which Pcdh15-CD2 is lost after hair-bundle development, in contrast to KO Pcdh15Δ'CD2 mice that lack Pcdh15-CD2 throughout development. Scale bars: 2 μm.

E    Scanning electron micrographs showing IHC and OHC hair bundles of $Pcdh15^{ex38-fl/ex38-fl}$ (control) and conditional Pcdh15Δ'CD2 mice on P30 and P45. White arrowheads indicate stereocilia with prolate-shaped tips, and arrows show regression of small row stereocilia. Tip-links (black arrowheads) are visible in controls on P30 and P45 but not in conditional Pcdh15ΔCD2 mice at either age. Scale bars: 500 nm.

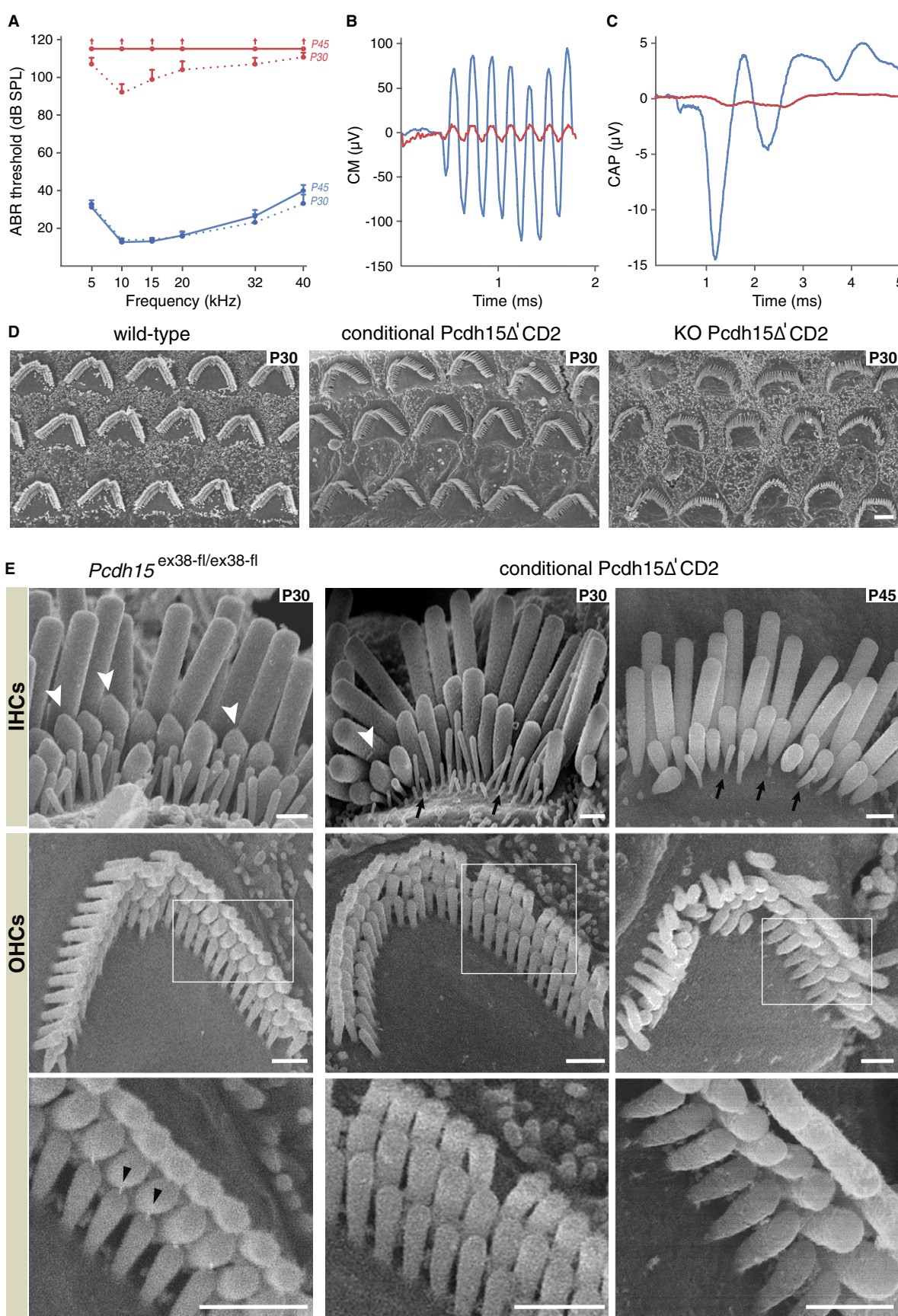

**Figure 2.**

truncated form (lacking the 273 C-terminal amino acids) or to the absence of the CD2 isoform due to nonsense-mediated mRNA decay (Maquat, 1995). The patient from family CPID4744 carried this mutation in the homozygous state, and the two patients from family CPIDS6-10 carried it in the heterozygous state, in association with a nonsense mutation, c.400C>T (p.R134*), located in *PCDH15* exon 6, which is common to the three Pcdh15 isoforms. Segregation analysis of these mutations (see Methods) showed that each parent had transmitted one mutation to the affected children, who are thus compound heterozygotes. Five siblings of the two families (patient IV.1 from family CPID4744 and patients IV.1, IV.2, IV.3 and IV.6 from family CPIDS6-10) with normal hearing did not carry the c.4542dup (p.P1515Tfs*4) mutation in the homozygous state (family CPID4744) or in the compound heterozygous state (family CPIDS6-10). Whole exome sequencing was performed in the three patients and did not detect any mutations with predicted pathogenicity in either the homozygous or the compound heterozygous state in known deafness genes or in other genes.

These three patients only show hearing impairment and no clinical signs of vestibular dysfunction (no delay in walking and no video-nystagmography abnormalities) or retinal defects (no vision difficulties in reduced illumination, and no abnormalities in any of fundus autofluorescence, optical coherence tomography, or scotopic or photopic electroretinogram; Fig 3). Abnormalities of the electroretinogram are systematically detected earlier in Usher 1 patients. This therefore suggests that the absence of the Pcdh15-CD2 isoform is responsible for an isolated (nonsyndromic) form of profound deafness.

# Conclusion

The functional and morphological defects of conditional Pcdh15Δ'CD2 mice show that Pcdh15-CD2, an isoform with a specific 284-amino acid C-terminal region, is an essential component of the tip-link in mature auditory hair cells. Knockout mice for Pcdh15-CD1 or Pcdh15-CD3 are not hearing-impaired (Webb *et al*, 2011), and consequently, we can conclude that the Pcdh15-CD2 isoform is the only Pcdh15 isoform required for the maintenance and/or function of the MET machinery in the two types of mature auditory sensory cells, the IHCs and OHCs. Pcdh15-CD2 is therefore essential both for hair-bundle morphogenesis, as a component of the kinociliary links, and later in mature auditory hair cells, for MET, as a component of the tip-links. Our work thus demonstrates that the auditory MET machinery undergoes a molecular maturation process, switching from a developmental form in which the Pcdh15 isoforms are functionally redundant to a fully mature form in which Pcdh15-CD2 is critical. Identifying which physiological features of the MET machinery are modified by this molecular maturation would require *ex vivo* MET current recording which, however, has not yet been successfully implemented at mature stages.

Our results should lead to a shift of focus in the search for new components of the lower part of the MET machinery. Because of the apparent functional redundancy observed between the various Pcdh15 isoforms, this search has so far been concentrated on proteins that can interact within the stereocilia with the sequences common to the three Pcdh15 isoforms, that is, the transmembrane and the juxtamembrane sequences (a total of 81 amino acids). Our

findings indicate that the identification of the ligands of the Pcdh15-CD2 cytoplasmic C-terminal region may be particularly pertinent. The absence of Pcdh15-CD2 affects mature auditory transduction but not vestibular transduction, both in patients and in the conditional knockout mice, showing that the function of Pcdh15-CD2 in the inner ear is conserved between mice and humans. By demonstrating the requirement for the Pcdh15-CD2 isoform for auditory function in humans, this constitutes a major step towards the development of gene therapy strategies for deafness caused by *PCDH15* defects.

# Materials and Methods

### Animals

Animals were housed in the Institut Pasteur animal facilities accredited by the French Ministry of Agriculture to perform experiments on live mice (accreditation 75-15-01, issued on 6 September 2013 in appliance of the French and European regulations on care and protection of the Laboratory Animals (EC Directive 2010/63, French Law 2013-118, 6 February 2013). The corresponding author confirms that protocols were approved by the veterinary staff of the Institut Pasteur animal facility and were performed in compliance with the NIH Animal Welfare Insurance #A5476-01 issued on 31 July 2012.

### *Pcdh15*[av3J/av3J] mice

*Pcdh15*[av3J/av3J] mice were obtained from Jackson Laboratories (Bar Arbor, ME).

### *Pcdh15* [ex38-fl/ex38-fl] mice

A targeting vector was designed in which loxP sites were introduced upstream and downstream from *Pcdh15* exon 38, and a neomycin resistance (neo) cassette flanked with Frt sites as selectable marker was introduced downstream of exon 38. The targeting construct was introduced by electroporation into embryonic stem (ES) cells from the 129S1/SvlmJ mouse strain, and positive ES cells were selected by their resistance to G418. Stem cells carrying the intended construct were injected into blastocysts from C57BL/6J mice to obtain chimeric mice. After germline transmission, mice were crossed with C57BL/6J mice producing Flp recombinase to remove the neo cassette. The *Pcdh15*[ex38-fl/ex38-fl] mice (MGI: 5566900) lack the neo cassette and behave like wild-type (*Pcdh15*[+/+]) mice. *Pcdh15* [ex38-fl/ex38-fl] mice were crossed with *PGK-cre* transgenic mice carrying the *cre* recombinase gene driven by the early and ubiquitously active phosphoglycerate kinase-1 gene promoter (Lallemand *et al*, 1998; mutant offspring referred to as KO Pcdh15Δ'CD2 mice); they were also crossed with Myo15-*cre* recombinant mice carrying the *cre* recombinase gene driven by the myosin-15 gene promoter which, in the inner ear, deletes the floxed fragment only in hair cells and after the period of hair-bundle development (Caberlotto *et al*, 2011; mutant offspring referred to as conditional Pcdh15Δ'CD2 mice).

The genotype of mice recombinant for *Pcdh15* exon 38 was verified by two PCR amplifications: one using oligo-Lf5707 (5'-cctccacgaaataacagtttctgtagc-3') and oligo-Er5711 (5'- cacatccatg-

    

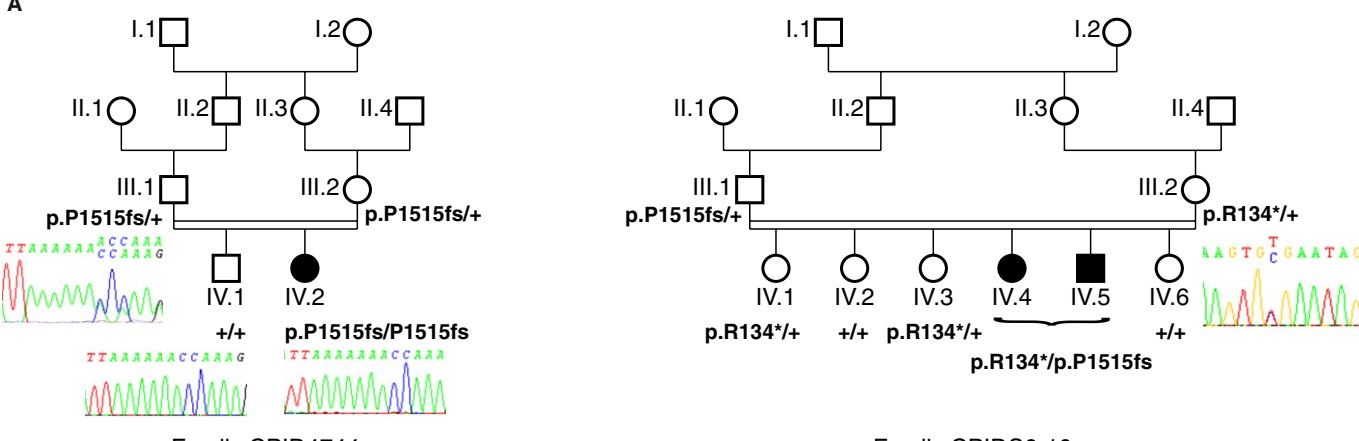

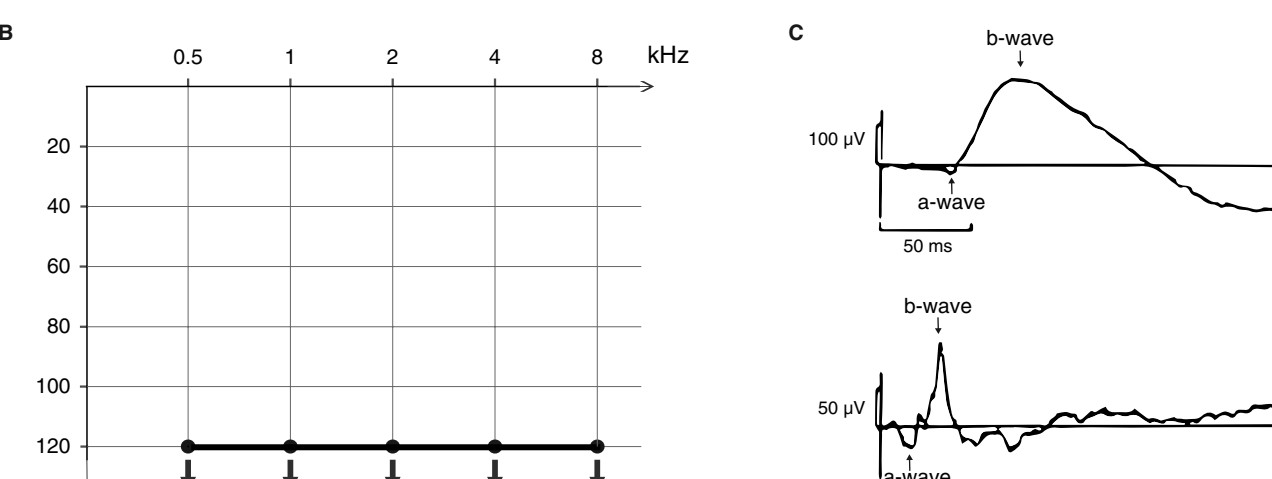

**Figure 3.  Profound deafness in patients carrying mutations affecting PCDH15-CD2.**

A   Segregation of the *PCDH15* mutations in the two families.

B   Air-conduction pure-tone audiometric curves for patients IV.2 (family CPID4744) and IV.4 (family CPIDS6-10) at the age of 3 and 5 years, respectively. Hearing thresholds at all sound frequencies tested (0.5, 1, 2, 4 and 8 kHz) were above 120 dB HL, the largest intensity tested, for both ears in both patients, indicating bilateral profound deafness.

C   Scotopic and photopic electroretinogram in patient IV.2 (family CPID4744) at the age of 7 years, showing normal a- and b-waves in both traces.

taactctagtctgtaac-3′) to detect the wild-type (2201-bp amplicon), the floxed (2386-bp amplicon) and the deleted (415-bp amplicon) alleles; and one using oligo-Ef5709 (5′- ccccagtgttgcttaagttttgcaac-3′) and oligo-Er5710 (5′- cagcagttgaagcatcatggtgttctg-3′) to detect an allele lacking *Pcdh15* exon 38. All studies were performed on mixed C57BL/6–129/Sv genetic backgrounds.

**Anti-Pcdh15-CD2 antibody and immunofluorescence experiments**

A rabbit polyclonal antibody was produced against the C-terminal part of Pcdh15-CD2 (aa1652-1790, GenBank Accession No. Q0ZM28) encoded by exon 38, specific to this isoform. The antigen coupled to an NHS column (GE Healthcare) was used for affinity purification of this antibody from the immune serum.

The affinity-purified antibody was used for immunofluorescence experiments on whole-mount preparations of the murine organ of Corti. After dissection, the tissue was fixed in 4% paraformaldehyde in phosphate-buffered saline (PBS) for 1 h at room temperature, incubated for 1 h at room temperature in PBS containing 20% normal goat serum and 0.3% Triton X-100 and incubated overnight with the primary antibody in PBS containing 1% bovine serum albumin (BSA). The secondary antibody was ATTO 488-conjugated goat anti-rabbit IgG antibody (Sigma-Aldrich, 1:200 dilution). Actin was labelled with ATTO 565-conjugated phalloidin (Sigma-Aldrich, 1:500 dilution). Samples were mounted in Fluorsave (Calbiochem, USA). The z-stack images were captured with a 63× Plan Apochromat oil immersion lens (NA 1.4) using a Zeiss LSM-700 confocal microscope and processed using Zeiss LSM image browser.

## Immunoblotting

Recombinant modified pcDNA3 vectors encoding the entire cytoplasmic regions of mouse cadherin-23, Pcdh15-CD2 and Pcdh15-CD3 (GenBank Accession Nos. Q99PF4, Q0ZM28, and Q0ZM20, respectively) coupled to an N-terminal Flag-tag were constructed for expression in HEK-293 cells. Cell lysates were prepared in Nupage Sample Buffer (Invitrogen).

A recombinant modified pFastbac vector encoding the cytoplasmic domain of mouse Pcdh15-CD1 (GenBank Accession No. NP_001165405) was constructed for expression in Sf9 insect cells. The N-terminally tagged fusion product, 6His-Flag-protein, was purified on a Ni-NTA column.

All samples were resolved by 4–8% Nupage SDSPAGE (Invitrogen). Proteins were transferred to PVDF membranes (Millipore) and immunoprobed, and bound antibody was detected by enhanced chemiluminescence (Pierce Biotechnology). A monoclonal anti-flag antibody was used to detect the four recombinant proteins (M2 Sigma-Aldrich, 1:500 dilution).

## Transmission electron microscopy

For immunogold labelling, cochleas were fixed overnight in 1% paraformaldehyde in PBS at 4°C. Samples were then washed three times in PBS, and cochlear coils dissected and transferred to PBS containing 10% normal horse serum and 0.1% Triton X-100 for 1 h at room temperature; they were then incubated overnight at 4°C in the same solution containing affinity-purified antibody diluted to 1:100. Samples were washed three times in PBS and post-fixed in 1% paraformaldehyde for 10 min at room temperature, then washed again in PBS and incubated overnight at 4°C with 5 nm gold-conjugated goat anti-rabbit IgG diluted 1:10 in PBS containing 10% horse serum, 0.05% Tween-20 and 1 mM EDTA. After extensive washing in PBS, samples were fixed for two hours at room temperature with 2.5% glutaraldehyde in 0.1 M sodium cacodylate, pH 7.4, containing 1% tannic acid, washed three times in 0.1 M sodium cacodylate buffer and then post-fixed for 1 h in 1% osmium tetroxide in 0.1 M sodium cacodylate buffer. The samples were further washed in 0.1 M sodium cacodylate buffer, and briefly in water, dehydrated through increasing concentrations of ethanol and embedded in TAAB 812 resin. Thin sections were cut with a diamond knife, mounted on copper mesh grids, double stained with uranyl acetate and lead citrate, and viewed in a Hitachi 7100 electron microscope operating at 100 kV. Images were captured with a Gatan Ultrascan 1000 camera at $2,048 \times 2,048$ pixel resolution.

## Audiological tests on recombinant mice

Mice were anaesthetised with xylazine and ketamine. Auditory brainstem responses (ABRs) were recorded through electrodes placed at the vertex and ipsilateral mastoid, with the lower back as the earth. Pure-tone stimuli at 5, 10, 15, 20, 32 and 40 kHz were used. Sound levels between 15 dB and 115 dB SPL in 5 dB steps were tested. ABR thresholds were determined as the lowest stimulus level resulting in recognizable waves. The compound action potential (CAP) and cochlear microphonic (CM) responses to a 5 kHz pure-tone stimulus at 105 dB SPL were collected between an electrode inserted in the round window and the vertex. The response

from the electrodes was amplified (gain 10,000), filtered, digitally converted and averaged using a comprised-data acquisition system. Distortion-product otoacoustic emissions (DPOAEs) were collected in the ear canal using a microphone. Two simultaneous pure-tone stimuli, at frequencies $f_1$ and $f_2$, were used with the same levels, from 30 to 75 dB SPL in 5 dB steps. The $f_2$ frequency was swept from 5 to 20 kHz in 1/8th octave steps, with $f_1$ chosen such that the frequency ratio $f_2/f_1$ was 1.20. Only the cubic difference tone at $2f_1–f_2$, the most prominent one from the ear, was measured (Le Calvez *et al*, 1998). Statistical significance was tested by the two-tailed unpaired t-test with Welch's correction.

## Vestibular tests on recombinant mice

The trunk curl test, the contact righting test and the swim test were carried out as previously described to analyse vestibular function (Hardisty-Hughes *et al*, 2010).

## Scanning electron microscopy

Organs of Corti were fixed for 1 h in 2.5% glutaraldehyde in 0.1 M aqueous sodium cacodylate solution at room temperature, followed by alternating incubations in 1% osmium tetroxide and 0.1 M thiocarbohydrazide (OTOTO). After dehydration by increasing concentrations of ethanol and critical point drying, samples were analysed by field emission scanning electron microscopy with a Jeol JSM6700F operating at 3 kV.

## Patients

This study was approved by the Local Ethical Committees and the Committee for the Protection of Individuals in Biochemical Research as required by French legislation. Written consent for genetic testing was obtained from all family members. Experiments conformed to the principles set out in the WMA Declaration of Helsinki and the NIH Belmont Report.

## Auditory tests on patients and family members

All family members underwent pure-tone audiometry in a soundproof room, with recording of air-conduction and bone-conduction thresholds. The air-conduction pure-tone average (ACPTA) threshold at frequencies 0.5, 1, 2 and 4 kHz was measured for each ear, and the value for the best ear was used to define the severity of deafness.

## Ophthalmological tests on patients

Electroretinograms were performed using Moncolor cupolas according to the International Society for Clinical Electrophysiology of Vision (ISCEV) protocol.

## DNA sequencing by the Sanger technique

Genomic DNA was extracted from peripheral blood by standard procedures. The 40 coding exons and flanking intronic sequences of *PCDH15* were amplified by PCR (primer sequences and conditions available upon request).

## Whole exome sequencing

Genomic DNA was captured using the Agilent enrichment solution method (SureSelect Human All Exon Kit Version 2, Agilent) with the Agilent bank of biotinylated oligonucleotide probes (Human All Exon v2 - 50 Mb, Agilent), followed by a high-throughput sequencing of the 75 bases at each end on an Illumina HiSeq 2000 (Gnirke *et al*, 2009). The sequence captures, enrichment and elution were performed according to the supplier's protocol and recommendations (SureSelect, Agilent) without modification. Briefly, 3 μg of each genomic DNA was fragmented by sonication, and fragments of 150–200 bp were purified. The oligonucleotide adapters for sequencing two ends of the fragments were ligated and repaired with an adenine added at the ends and then purified and enriched by 4–6 PCR cycles. Aliquots of 500 ng of these purified libraries were then hybridized to SureSelect bank capture oligonucleotide probes for 24 h. After hybridization, washing and elution, the eluted fraction was amplified by 10–12 cycles of PCR to obtain sufficient DNA template for further downstream processes, purified and quantified by quantitative PCR. Each DNA sample was then eluted, enriched and the 75-base sequences from each end determined on an Illumina HiSeq 2000. The Illumina pipeline RTA version 1.14 with default settings was used for image analysis and sequence determination.

## Bioinformatics analysis

The CASAVA1.8 pipeline provided by Illumina was used for bioinformatics analysis of sequence data. CASAVA1.8 is a suite of scripts including sequence alignment of the complete genome (build37), counting and detection of allelic variants (SNPs and Indels). The alignment algorithm used was ELANDv2e (Maloney alignment and multi-seed mismatch reducing artefact). The annotation of genetic variation was made internally, including the annotation of genes (RefSeq) and referenced polymorphisms (HapMap, 1000Genomes and Exome Variant Server) followed by characterization of the variation (exonic, intronic, silent, missense, etc.) as previously described (Delmaghani *et al*, 2012).

**Supplementary information** for this article is available online: http://embomolmed.embopress.org

## Author contributions

EP generated CD2-antibodies and performed the immunoblotting experiments. EP and VM performed immunolabelling experiments, VM performed hair-bundle SEM, and both EP and VM analysed the results. RG performed TEM immunogold-labelling experiments, PA performed auditory testing. SG, MH, SM, MM and AZ contributed to clinical and genetic evaluation of the patients. CB and SA analysed results of whole exome sequencing and mutations in *PCDH*5 in the two family members. TD provided and genotyped all the recombinant mice used in this work. EP, VM, CB, JPH, GR, PA, AB and CP wrote the article. AB trained EP for biochemical studies and supervised her work. CP designed the whole project.

## Acknowledgements

The authors thank the family members for their participation in this study. We thank Aziz El-Amraoui for his comments and suggestions about the images. We are grateful to Dominique Weil and the Institut Clinique de la Souris (Illkirch, France) for producing Pcdh15-exon38 recombinant mice. We thank Anne Dieux, Yahia Rous, Hayet Lebdi and Ahmed Cheknane for the clinical examination of hearing-impaired individuals and their family members, and Isabelle Drumare-Bouvet and Kamel Boudjelida for the ophthalmological examinations. Furthermore, we want to show our gratitude towards Luce Smagghe, Asma Behlouli and Sylvie Nouaille for technical assistance. EP was supported by a fellowship from the Fondation Raymonde et Guy Strittmatter. This work was supported by ERC-Hair bundle (ERC-2011-ADG_294570), Foundation BNP Paribas and LHW-Stiftung to CP; Tassili project funding to CP and AZ and Wellcome Trust programme grant (WT087377) to GR. This work performed in the frame of the LABEX LIFESENSES [reference ANR-10-LABX-65] was supported by French state funds managed by the ANR within the Investissements d'Avenir programme under reference ANR-11-IDEX-0004-02.

## Conflict of interest

The authors declare that they have no conflict of interest.

---

### The paper explained

#### Problem

In the sensory hair cells of the ear, the mechano-electrical transduction machinery transforms sound wave energy into electrical signals. Elucidation of the molecular composition of this machinery is of substantial importance to understand the way it works. A fibrous link, called the tip-link, is an essential component of this machinery as it is necessary to open the mechano-electrical transduction ion channel and to activate the resulting sound processing cascade. The aim of this study was to identify which isoform(s) of protocadherin-15, the protein that makes up the lower part of the tip-link, form(s) this link in mature hair cells.

#### Results

Our work demonstrates that in mature auditory cells, the CD2 isoform of protocadherin-15 is an essential component of the tip-links. Its absence from mouse hair cells results in the loss of the mechano-electrical transduction process, hence in profound deafness; we also provide genetic evidence that this conclusion also applies to humans. It has been reported that the three different isoforms of protocadherin-15 are functionally redundant in the mouse cochlea at immature stages (during the formation of the auditory sensory cells, when mice do not yet hear although the mechano-electrical transduction can operate). We show that this is not the case at mature stages (after the onset of hearing).

#### Impact

Gene therapy for Usher syndrome type 1 caused by mutations in the protocadherin-15 (*PCDH15*) gene is particularly challenging because this gene encodes three isoforms. The isoform(s) to be delivered to prevent or cure the hearing and the retinal defects involved in this syndrome need(s) to be identified. Our work, by demonstrating that the protocadherin-15-CD2 isoform is essential for auditory function but not visual function both in mice and humans, provides key information. The mouse mutants are therefore suitable for use as animal models for the development of gene therapy for deafness caused by *PCDH*15 defects.

---

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
