## [Review Process File · EMBO Molecular Medicine]

The CD2 isoform of protocadherin-15 is an essential component of the tip-link complex in mature auditory hair cells

Elise Pepermans, Vincent Michel, Richard Goodyear, Crystel Bonnet, Samia Abdi, Typhaine Dupont, Souad Gherbi, Muriel Holder, Mohamed Makrelouf, Jean-Pierre Hardelin, Sandrine Marlin, Akila Zenati, Guy Richardson, Paul Avan, Amel Bahloul and Christine Petit

Corresponding author: Christine Petit, Institut Pasteur

Review timeline:

Submission date:	14 February 2014
Editorial Decision:	19 March 2014
Revision received:	02 May 2014
Editorial Decision:	08 May 2014
Accepted:	08 May 2014

Transaction Report:

Editor: Céline Carret

1st Editorial Decision

19 March 2014

Thank you for the submission of your manuscript to EMBO Molecular Medicine. We have now heard back from the three referees whom we asked to evaluate your manuscript.

As you will see from the enclosed reports, the 3 referees are overall supportive of publication pending some clarifications (ref. 2 and 3) and biochemical evidence to show the loss of CD2 (ref. 3).

We would welcome the submission of a revised version of your manuscript and depending on the nature of the revisions, this may be sent back to the referees for another round of review. In addition, and in order to gain time, I would appreciate if you could provide editorial amendments [...].

Please note that it is EMBO Molecular Medicine policy to allow a single round of revision in order to avoid the delayed publication of research findings. Consequently, acceptance or rejection of the manuscript will depend on the completeness of your responses included in the next version of the manuscript.

I look forward to seeing a revised form of your manuscript.

***** Reviewer's comments *****

Referee #1 (Remarks):

One of the questions regarding the mechanotransduction of the hair cell is which form of Pcdh15 is essential for this function. In order to properly target this region in cases of Pcdh15 deafness, the identification of the isoform is required. The authors set out to determine which of the 3 isoforms is the essential component. They employed the use of an elegant mouse model that only had the CD2 isoform missing. Most compelling, they were able to extend this finding to humans, since they identified a Pcdh15 mutation affecting only this isoform.

The first goal was to determine the localization of the CD2 isoform, which they did by developing a specific antibody (specificity validated and included in supp data). They developed a mouse model, using the Myo15a-cre driven promoter to remove Pcdh15-CD2 from hair cells. The auditory function and morphological characteristics were examined well, reported both in the regular and supp data. Finally, the human patients were identified and linked to the findings in the mouse. They confirmed that there were no additional mutations in the patients by using exome sequencing. The authors suggest that the identification of the essential role of this isoform will be beneficial for gene therapy.

Referee #2 (Remarks):

The paper by Pepermans et al. reports a beautiful set of experiments that throws new light on the role of the CD2 isoform of Pcdh15 in deafness. It had previously been reported [Webb et al.] that a constitutive knockout of the CD2 isoform led to planar cell polarity defects and deafness, while the CD1 and CD3 isoform knockouts were normal. However, tip link and (mechano-electrical transducer) MET currents were reported to be normal in the CD2 knockout mice suggesting that there was functional redundancy between the various isoforms. The authors have however questioned this conclusion, and proceeded to analyse a conditional mutant for CD2 and show in mature hair cells from P17 onwards, that ABR thresholds are affected and that by P30 tip links are missing and MET currents are profoundly affected. The authors' data is compelling. They observe the planar cell polarity phenotype found in the constitutive knockout, but their new data clearly shows that in the mature hair cell there is no functional redundancy between the isoforms, and that the CD2 isoform is clearly required for the formation of tip links and the generation of MET currents. This is important new information on the role of Pcdh15 in the auditory transduction machinery.

Interestingly, they have identified human deafness patients with no evidence for retinal dysfunction that carry CD2 specific mutations. This confirms the authors' hypothesis of the specific role of this isoform and its involvement in profound, non-syndromic deafness with implications for gene therapy.

The paper is carefully written, and I have no major comments for revision. However, the authors might consider elaborating on two points:

1. The authors point out in the legend to Fig. 1 that Pcdh15-CD2 is restricted to the tip of the short and middle row stereocilia, and the apico-lateral region of the tallest stereocilia. How does they reconcile this with their conclusion in the text that CD2 is a lower tip-link component? It would be useful in this section for the authors to clarify their conclusions about the labeling in the tallest stereocilia row.
2. The authors show at P30 a constitutive knockout demonstrating the planar cell polarity defect. However, presumably the constitutive knockout, like the conditional, would be expected to show absence of tip links and MET currents, from P30 and beyond. The authors should comment on this. The failure of Webb et al. to observe this effect presumably reflects the fact that their observations were confined to around P7.

Referee #3 (Comments on Novelty/Model System):

The authors develop a novel model (developmentally controlled conditional mouse model). The usefulness of the model is nicely supported by finding the related genetic condition in human hereditary deafness

Referee #3 (Remarks):

Here, Pepermans et al. show that of the three splice isoforms of protocadherin-15, Pcdh-CD2 is the one that contributes to functional tip-links in mature mammalian auditory hair cells (but not vestibular hair cells).

First, by generating a novel isoform-specific antibody, Pcdh-CD2 is shown in mature stereociliary tips, contrary to previous believe. Second, by generating a conditional postnatal and isoform specific knockout, it is shown that independent of a previously acknowledged role in hair cell development, the CD2 isoform is absolutely required for mechano-electrical transduction (MET), and for the presence of tip links in adult inner and outer hair cells. The conclusions on MET function are based on a battery of systems measurements since direct measurement of MET in adult mouse hair cells is difficult. Third, a human frameshift mutation that selectively deletes the CD2 isoform is shown to cause non-syndromic deafness.

Besides advancing our understanding of the composition of the MET machinery, the findings are relevant for two main reasons: the lower end of the MET machinery contains other components, including the MET channel, but the composition of this complex is not well understood. Knowledge of the intracellular domain of Pcdh-15 will open new strategies to elucidate the composition and interactions within this complex. Second, knowledge of the relevant isoform will be necessary for future gene therapy of Usher syndrome where protocadherin-15 (all isoforms) is affected.

This manuscript is well written, the data are of high quality and the conclusions appear sound.

There is one issue that should be addressed to improve the manuscript.

In the conditional Pcdh15-deltaCD2 mice, the developmental loss of hearing seems entirely consistent with a gradual depletion of the CD2 isoform. However I am missing information on the actual loss of the protein from the hair cells. First, what is the expected time course of the postnatal onset of Cre expression by the Myo15-Cre driver mice? Second, the protein expression levels should be documented throughout postnatal development. The obviously very useful CD2-specific antibody should allow for these experiments either by immunohistochemistry or by western blots (as in Fig. 1b, Suppl. Fig1). Also, does the expression of the CD1 and CD 3 isoforms stay constant?

Minor points

1. It is stressed throughout the manuscript that CD2 isoform becomes essential in mature hair cells. The meaning of mature needs to be clear. E.g., cells in Fig. 1 include rather early (P5) postnatal hair cells, but also P16.
2. Although it eventually becomes clear, it would be helpful to explicitly state in the abstract and introduction that the three isoforms are splice isoforms.
3. It would be reassuring to have more quantitative data on the immuno-gold labeling.

1st Revision - authors' response

02 May 2014

We thank the reviewers for their comments and suggestions.

***** Reviewer's comments *****

Referee #1 (Remarks):

One of the questions regarding the mechanotransduction of the hair cell is which form of Pcdh15 is essential for this function. In order to properly target this region in cases of Pcdh15 deafness, the

identification of the isoform is required. The authors set out to determine which of the 3 isoforms is the essential component. They employed the use of an elegant mouse model that only had the CD2 isoform missing. Most compelling, they were able to extend this finding to humans, since they identified a *Pcdh15* mutation affecting only this isoform.

The first goal was to determine the localization of the CD2 isoform, which they did by developing a specific antibody (specificity validated and included in supp data). They developed a mouse model, using the *Myo15a-cre* driven promoter to remove *Pcdh15-CD2* from hair cells. The auditory function and morphological characteristics were examined well, reported both in the regular and supp data. Finally, the human patients were identified and linked to the findings in the mouse. They confirmed that there were no additional mutations in the patients by using exome sequencing. The authors suggest that the identification of the essential role of this isoform will be beneficial for gene therapy.

Referee #2 (Remarks):

The paper by Pepermans et al. reports a beautiful set of experiments that throws new light on the role of the CD2 isoform of *Pcdh15* in deafness. It had previously been reported [Webb et al.] that a constitutive knockout of the CD2 isoform led to planar cell polarity defects and deafness, while the CD1 and CD3 isoform knockouts were normal. However, tip link and (mechano-electrical transducer) MET currents were reported to be normal in the CD2 knockout mice suggesting that there was functional redundancy between the various isoforms. The authors have however questioned this conclusion, and proceeded to analyse a conditional mutant for CD2 and show in mature hair cells from P17 onwards, that ABR thresholds are affected and that by P30 tip links are missing and MET currents are profoundly affected. The authors' data is compelling. They observe the planar cell polarity phenotype found in the constitutive knockout, but their new data clearly shows that in the mature hair cell there is no functional redundancy between the isoforms, and that the CD2 isoform is clearly required for the formation of tip links and the generation of MET currents. This is important new information on the role of *Pcdh15* in the auditory transduction machinery.

Interestingly, they have identified human deafness patients with no evidence for retinal dysfunction that carry CD2 specific mutations. This confirms the authors' hypothesis of the specific role of this isoform and its involvement in profound, non-syndromic deafness with implications for gene therapy.

The paper is carefully written, and I have no major comments for revision. However, the authors might consider elaborating on two points:

1. The authors point out in the legend to Fig. 1 that *Pcdh15-CD2* is restricted to the tip of the short and middle row stereocilia, and the apico-lateral region of the tallest stereocilia. How does they reconcile this with their conclusion in the text that CD2 is a lower tip-link component? It would be useful in this section for the authors to clarify their conclusions about the labeling in the tallest stereocilia row.

The apico-lateral labelling of the tallest stereociliary row in mature OHCs and IHCs may be associated to intrarow interstereociliary fibrous links. In mice, such apical links have been reported both in IHCs and OHCs during their maturation, but at a mature stage in OHCs only. However, in guinea pigs, these links have also been reported in mature IHCs (Furness & Hackney, 1985), suggesting that they may have been overlooked in adult mice. Although we have obtained preliminary information in support of this hypothesis, firm conclusion regarding the interpretation of the labelling observed in the tallest stereocilia row will rely on an entirely new set of data that, we believe, is beyond the scope of the present report. However, we mentioned in the revised version of the manuscript that this labelling may be associated with intrarow interstereociliary fibrous links (see legend to figure 1, page 16).

(c) From left to right: Transmission electron micrographs of a *Pcdh15-CD2* immunoreactive IHC hair bundle at P15 (left panel), enlargements of boxed region: two adjacent sections of the same hair

bundle (central panels), and a tip-link profile from an OHC (right panel). Arrowheads indicate gold particles.

Note that the presence of the gold particles is restricted to the tips of the short and middle row stereocilia, consistent with Pcdh15-CD2 being a component of the lower part of the tip-link, and to the apico-lateral region of the tallest stereocilia, suggesting that Pcdh15-CD2 could also be a component of the lateral links between stereocilia of the tallest row.

2. The authors show at P30 a constitutive knockout demonstrating the planar cell polarity defect. However, presumably the constitutive knockout, like the conditional, would be expected to show absence of tip links and MET currents, from P30 and beyond. The authors should comment on this. The failure of Webb et al. to observe this effect presumably reflects the fact that their observations were confined to around P7.

The paper by Webb et al. showed the presence of tip-links in *PCDH15-ΔCD2* mice at P1, by using high magnification scanning EM. In contrast, at mature stage (P21 and P96) the low magnification scanning EM used could not enable tip-link visualisation.

In this revised version of the manuscript, we added a figure (Supplementary Fig 6) showing the morphology of IHC and OHC hair bundles in our *Pcdh15Δ^{CD2}* knock-out (*PGK-Cre*) mice at P30. As expected, in the hair bundles of these mutant mice, the absence of tip-links and a regression of the short and middle stereociliary rows are observed.

Referee #3 (Comments on Novelty/Model System):

The authors develop a novel model (developmentally controlled conditional mouse model). The usefulness of the model is nicely supported by finding the related genetic condition in human hereditary deafness

Here, Pepermans et al. show that of the three splice isoforms of protocadherin-15, Pcdh-CD2 is the one that contributes to functional tip-links in mature mammalian auditory hair cells (but not vestibular hair cells).

First, by generating a novel isoform-specific antibody, Pcdh-CD2 is shown in mature stereociliary tips, contrary to previous believe. Second, by generating a conditional postnatal and isoform specific knockout, it is shown that independent of a previously acknowledged role in hair cell development, the CD2 isoform is absolutely required for mechano-electrical transduction (MET), and for the presence of tip links in adult inner and outer hair cells. The conclusions on MET function are based on a battery of systems measurements since direct measurement of MET in adult mouse hair cells is difficult. Third, a human frameshift mutation that selectively deletes the CD2 isoform is shown to cause non-syndromic deafness.

Besides advancing our understanding of the composition of the MET machinery, the findings are relevant for two main reasons: the lower end of the MET machinery contains other components, including the MET channel, but the composition of this complex is not well understood. Knowledge of the intracellular domain of Pcdh-15 will open new strategies to elucidate the composition and interactions within this complex. Second, knowledge of the relevant isoform will be necessary for future gene therapy of Usher syndrome where protocadherin-15 (all isoforms) is affected.

This manuscript is well written, the data are of high quality and the conclusions appear sound.

Referee #3 (Remarks):

There is one issue that should be addressed to improve the manuscript.

In the conditional Pcdh15-deltaCD2 mice, the developmental loss of hearing seems entirely consistent with a gradual depletion of the CD2 isoform. However I am missing information on the actual loss of the protein from the hair cells. First, what is the expected time course of the postnatal onset of Cre expression by the Myo15-Cre driver mice? Second, the protein expression levels

should be documented throughout postnatal development. The obviously very useful CD2-specific antibody should allow for these experiments either by immunohistochemistry or by western blots (as in Fig. 1b, Suppl. Fig1). Also, does the expression of the CD1 and CD 3 isoforms stay constant?

By using a reporter gene, we previously showed that the recombination induced by the Cre recombinase driven by the *Myo15* promoter takes place only in hair cells. It starts on P0 and extends over the entire cochlea on P4; it starts on E19 in the vestibule. For every inactivated gene, the complete loss of its encoded protein is expected to depend on the spatio-temporal distribution of the recombination events, the lifetime of the corresponding transcript and the turnover of the protein. These are variable from one gene to another. For instance in previous studies, conditional knock out of the *ans* encoding gene using the same recombinant *Myo15*-Cre mice, resulted in the disappearance of the mechano-transduction process already at P9 whilst conditional knock out of cadherin-23 resulted in its disappearance only at P21 (onset of hearing loss) (Caberlotto et al, 2011).

In conditional *Pcdh15 Δ 'CD2* mice the loss of mechanotransduction only takes place from P21 onwards. This is a major obstacle for a direct visualization of the loss of the *Pcdh15*-CD2 isoform. Indeed, even though our antibody provides a correct and specific labelling of *Pcdh15*-CD2 up to P16, this staining is no longer detected thereafter. This is presumably due to the masking of the recognized epitope, a common situation observed with many other antibodies in mature hair bundles. Regarding the possible use of an alternative detection method, biochemical (western blot) analysis cannot be used due to the very small number of hair cells (a few thousands per cochlea); even the use of a very large number of mice will not solve this issue because *Pcdh15*-CD2 is also expressed in the supporting cells of the cochlea, and this expression is unchanged in the conditional *Pcdh15 Δ 'CD2* mice.

The detection of *Pcdh15*-CD2 mRNA resulting from the loss of exon 38 in conditional *Pcdh15 Δ 'CD2* mice encounters the same difficulties: this 3'-terminal exon of the spliced isoform being deleted, this transcript cannot be directly detected and again normal *Pcdh15*-CD2 transcripts are still produced by supporting cells.

The only possibility left is DNA analysis. We therefore extracted DNA from microdissected organs of Corti (cochlear sensory epithelium) and from the sensory areas of the vestibule (inner ear sample) in conditional *Pcdh15 Δ 'CD2* mice and *Pcdh15^{ex38-fl/ex38-fl}* mice, at P21. As a negative control we also extracted DNA from the tail and striated muscle tissues of these mutant mice, and from the tail of wild-type mice. Primers (see below) were designed to amplify each of the exons that characterize the various protocadherin15 isoforms (i.e. exon 35 for *Pcdh15*-CD1, exon 38 for *Pcdh15*-CD2, and exon 39 for *Pcdh15*-CD3). As shown below, in the inner ear (and not in the other tissues) of the conditional *Pcdh15 Δ 'CD2* mice, a smaller band could be detected for the amplicon of exon 38; its sequence demonstrated the deletion of this exon. The presence of a "not-recombined" amplicon (matching the size of the floxed exon) in the inner ear of conditional *Pcdh15 Δ 'CD2* mice is attributable to the presence of the supporting cells in these neuroepithelial extracts. Exon 35 and exon 39, specific to *Pcdh15*-CD1 and *Pcdh15*-CD3, respectively, are unaffected in the inner ear of conditional *Pcdh15 Δ 'CD2* mice, thus predicting a normal synthesis of these *Pcdh15* isoforms in the hair cells.

Exon 35 Forward: 5' - TCCCCAGAGTTACTGAGATGGGA-3'

Exon 35 Reverse: 5' - CAAGTCAACACGCCTTTGGT-3'

Amplicon WT: 2202 nucleotides

Exon 38 Forward: 5' - CCTCCACGAAATAACAGTTTCTGTAGC -3'

Exon 38 Reverse: 5' - CACATCCATGTAACCTAGTCTGTAAC -3'

Amplicon WT: 2201 nucleotides

Amplicon Flox: 2386 nucleotides

Amplicon del: 415 nucleotides

Exon 39 Forward: 5' - CTGGCTTTCTGCTGAACTTGCT -3'

Exon 39 Reverse: 5' - AGCATTGACCTCGAATGTTCTTCC -3'

Amplicon WT: 1500 nucleotides

Minor points

1. It is stressed throughout the manuscript that CD2 isoform becomes essential in mature hair cells. The meaning of mature needs to be clear. E.g., cells in Fig. 1 include rather early (P5) postnatal hair cells, but also P16.

This remark has been taken into account in the revised version of the manuscript (see legend to figure 1, page 16):

(b) Confocal images of hair cells (OHC above, IHC below) stained for Pcdh15-CD2 (green) and actin (red) at P5 (immature hair cells) and P16 (mature hair cells).

2. Although it eventually becomes clear, it would be helpful to explicitly state in the abstract and introduction that the three isoforms are splice isoforms.

This manuscript has been modified accordingly (page 2).

ABSTRACT

Protocadherin-15 (Pcdh15) is a component of the tip-links, the extracellular filaments that gate hair cell mechano-electrical transduction channels in the inner ear. There are three Pcdh15 splice isoforms (CD1, CD2, CD3), which only differ by their cytoplasmic domains; they are thought to function redundantly in mechano-electrical transduction during hair-bundle development, but whether any of these isoforms composes the tip-link in mature hair cells remains unknown.

INTRODUCTION

Three transmembrane protocadherin-15 (Pcdh15) splice isoforms (Pcdh15-CD1, Pcdh15-CD2, Pcdh15-CD3) differing only in the C-terminal part of their cytoplasmic domains (Fig 1a), are present in the hair-bundles of developing cochlear hair cells

3. *It would be reassuring to have more quantitative data on the immuno-gold labelling.*

This remark has been taken into account in the revised version of the manuscript. In the supplementary information section we added a figure (Supplementary Fig 3) showing the distribution of gold particles in 9 sections of mature inner hair cells (gold particles n=39) and showing the distribution and quantification (with histograms) in 16 sections of mature outer hair cells (n=120). These data show that the majority of gold particles are indeed present at the apex of the short and middle row stereocilia.

References:

Caberlotto E, Michel V, Foucher I, Bahloul A, Goodyear RJ, Pepermans E, Michalski N, Perfettini I, Alegria-Prevot O, Chardenoux S, Do Cruzeiro M, Hardelin JP, Richardson GP, Avan P, Weil D, Petit C (2011) Usher type 1G protein sans is a critical component of the tip-link complex, a structure controlling actin polymerization in stereocilia. *Proceedings of the National Academy of Sciences of the United States of America* 108: 5825-5830

Furness DN, Hackney CM (1985) Cross-links between stereocilia in the guinea pig cochlea. *Hearing research* 18: 177-188

2nd Editorial Decision

08 May 2014

Please find enclosed the final report on your manuscript. We are pleased to inform you that your manuscript is accepted for publication and is now being sent to our publisher to be included in the next available issue of EMBO Molecular Medicine.

***** Reviewer's comments *****

Referee #3 (Remarks):

The authors addressed all of my comments carefully and appropriately.
I have no further concerns.